# Bacterial Infection and Non-Hodgkin B-Cell Lymphoma: Interactions between Pathogen, Host and the Tumor Environment

**DOI:** 10.3390/ijms22147372

**Published:** 2021-07-09

**Authors:** Monika Maria Biernat, Tomasz Wróbel

**Affiliations:** Department of Haematology, Blood Neoplasms and Bone Marrow Transplantation, Wroclaw Medical University, 50-367 Wroclaw, Poland; tomasz.wrobel@umed.wroc.pl

**Keywords:** lymphomagenesis, bacteria, B-cell lymphoma, *Helicobacter pylori*, tumor environment, host factors

## Abstract

Non-Hodgkin B-cell lymphomas (NHL) are a heterogeneous group of lymphoid neoplasms with complex etiopathology, rich symptomatology, and a variety of clinical courses, therefore requiring different therapeutic approaches. The hypothesis that an infectious agent may initiate chronic inflammation and facilitate B lymphocyte transformation and lymphogenesis has been raised in recent years. Viruses, like EBV, HTLV-1, HIV, HCV and parasites, like *Plasmodium falciparum,* have been linked to the development of lymphomas. The association of chronic *Helicobacter pylori* (*H. pylori*) infection with mucosa-associated lymphoid tissue (MALT) lymphoma, *Borrelia burgdorferi* with cutaneous MALT lymphoma and *Chlamydophila psittaci* with ocular adnexal MALT lymphoma is well documented. Recent studies have indicated that other infectious agents may also be relevant in B-cell lymphogenesis such as *Coxiella burnettii*, *Campylobacter jejuni*, *Achromobacter xylosoxidans*, and *Escherichia coli.* The aim of the present review is to provide a summary of the current literature on infectious bacterial agents associated with B-cell NHL and to discuss its role in lymphogenesis, taking into account the interaction between infectious agents, host factors, and the tumor environment.

## 1. Introduction

Non-Hodgkin B-cell lymphomas (NHL) are the most common hematological malignancies worldwide and the fifth most common cancer. It is a heterogeneous group of lymphoid neoplasms, including latent types such as marginal-zone lymphomas (MZL), follicular lymphomas, and aggressive diseases such as diffuse large B-cell lymphoma (DLBCL) and Burkitt’s lymphoma, with complex etiopathology, a variety of symptomatology and clinical courses, and consequently different therapeutic approaches [1]. MZL are indolent diseases that arise from the small B cells of the marginal zone, which surrounds the lymph node and lies outside of the mantle zone. According to the latest WHO classification, there are three sub-types of MZL, namely extranodal MZL (EMZL), nodal MZL (NMZL) and splenic MZL (SMZL) [2,3]. Of the EMZLs, which are termed mucosa-associated lymphoma tissue (MALT) and account for 70% of MZLs, the most common lesions are located in the stomach (approximately 30–50%), with less frequent occurrences in the lung, skin, ocular adnexa, salivary glands, thyroid, breast and other sites (overall 8% of all NHLs). Furthermore, it is thought that autoimmune diseases such as Sjögren’s syndrome and Hashimoto’s disease may contribute to the development of MALT lymphomas, but the main driving factor has been attributed to chronic inflammatory processes [2,4,5]. For aggressive DLBCL and Burkitt lymphoma the epidemiology is different, and an intensive diagnostic and therapeutic approach should always be implemented as soon as possible.

Some specific bacterial species have been identified that correlate strongly with cancers. The microorganisms include *Salmonella typhimurium* which associates with hepatobiliary carcinoma, *Citrobacter rodentium* that can be a leading cause of human colorectal cancer, *Mycobacterium tuberculosis* which associates with lung cancer and two periopathogenic species *Fusobacterium nucleatum* and *Porphyromonas gingivalis* that play an important role in the development of colorectal and pancreatic cancer [6,7,8]. Some research also demonstrated the role of infections caused by bacteria, such as *Streptococcus* sp., *Peptostreptococcus* sp., *Prevotella* sp., *Fusobacterium* sp., *Porphyromonas gingivalis*, and *Capnocytophaga gingivalis* in the pathogenesis of oral cancer [8].

The role of infectious agents in the development of NHL has been studied for many years. The hypothesis that an infectious agent may initiate chronic inflammation, and therefore be the background of B lymphocyte transformation and lymphomagenesis has been raised in recent years [9,10,11]. Viruses, like EBV, HTLV-1, HIV, HCV and parasites, like *Plasmodium falciparum* have been linked to the development of lymphomas. Bacteria have been shown to be carcinogens and tumor promoters and can take part in tumorigenesis by activating various intracellular signaling pathways, modulating apoptosis and cell proliferation [12]. The complex relationship between these three components occurs when the cells of the immune system are at risk of malignant transformation, particularly in the case of lymphomas [13]. The association of chronic *Helicobacter pylori* (*H. pylori*) infection with MALT lymphoma, *Borrelia burgdorferi* with cutaneous MALT lymphoma and *Chlamydophila psittaci* with ocular adnexal MALT lymphoma is well documented [9,14]. Recent studies indicate that the list of infectious agents should be revised to include others that may also be relevant in B-cell lymphogenesis such as *Coxiella burnetti*, *Campylobacter jejuni*, *Achromobacter xylosoxidans*, and *Escherichia coli* [14]. Lymphogenesis is a complex process that can be influenced by many factors, not only tumor-dependent but also host-dependent, environmental, and epigenetic factors. Infectious agents can interact in complex ways with all these factors to promote tumor development and progression [15].

The aim of this review is to summarize the current literature on infectious bacterial agents associated with NHL, and to discuss their role in lymphogenesis taking into account the interaction between infectious agents and host factors, tumor environment and genetic factors.

## 2. Role of Bacteria in Gastrointestinal Lymphoma

### 2.1. H. pylori and MALT, DLBCL, and Burkitt lymphoma

*H. pylori* infection affects more than half of the world’s population. In approximately 10–20% of infected individuals, it manifests as chronic gastritis and ulcers, and in only less than 1% does it lead to gastric cancer or MALT lymphoma [16]. Although the role of *H. pylori* in the development of adenocarcinoma is well documented and the bacterium has been recognized as a class I carcinogen, it is still not entirely clear why only a small percentage of infected individuals trigger carcinogenesis and develop gastric cancer and MALT lymphomas [17,18]. Gastric MALT lymphoma is a low-grade NHL, and is known to be a consequence of chronic inflammatory processes induced by *H. pylori* [9,19]. Intensive research in recent years using modern molecular techniques along with cellular and animal models has demonstrated the importance of *H. pylori* as a carcinogen and has greatly expanded the knowledge of the interaction of these spiral bacteria on the lymphoproliferative processes [20,21,22]. The pathogenesis of *H. pylori*-dependent chronic gastritis leading to metaplasia and activation of the carcinogenic processes is still the subject of intense research. *H. pylori* has unique properties that allow it to colonize the human gastric mucosa for extended periods of time [23]. The most important virulence factors are the CagA protein and the VacA toxin [20]. The CagA protein induces epithelial cells to secrete interleukin 8 (IL-8) through activation of NF-κB [20,23]. CagA is then transported into gastric epithelial cells through the type IV secretion system (TFSS), where Src and Abl family kinases can phosphorylate the tyrosine residues of the EPIYA (Glu-Pro-Ile-Tyr-Ala) fragment of CagA. After phosphorylation, CagA binds to Src homology-2 domain phosphatase (SHP-2) and modulates multiple signaling pathways by activation of endoplasmic reticulum kinases (ERK), the p38 mitogen-activated protein kinase (MAPK), and an increase in the expression of Bcl-2 and Bcl-XL [24,25]. Activation of these signaling pathways causes changes in cell structure and architecture leading to reorganization of the actin cytoskeleton, cell elongation, disruption of B lymphocyte apoptosis and increased proliferation [20,25,26]. The CagA protein also affects the JAK-STAT signaling pathway through IL-3 secretion, which suppresses the proliferation of B-lymphocytes (Figure 1). However, the role of JAK-STAT signaling, as induced by CagA in the lymphogenesis of gastric MALT lymphoma, needs further research [26]. Interestingly, the role of CagA in lymphogenesis has been demonstrated in investigations examining tissue samples taken from patients suffering from gastric MALT lymphoma and DLBCL [27]. However, there is some debate surrounding whether the strains expressing the CagA protein are mainly associated with gastric DLBCL, and less frequently with gastric MALT lymphoma [28,29].

VacA cytotoxin can form intracellular vacuoles, leading to damage and disintegration of gastric epithelial cells. The activity of different VacA toxin alleles affects its cytotoxicity [30]. *H. pylori* strains housing the *vacAm2* allele were less biologically active in vivo, and less vacuolating in vitro; these strains are thought to be the predominant form in patients with gastric MALT lymphoma. Furthermore, the *H. pylori* strains possessing the *vacAs1m2* genotype associated with *iceA1* variants were found in MALT lymphoma patients at levels 5 times that of chronic gastritis [31]. Studies on overexpression of the p52 fragment of VacA toxin show that VacA induces the production of TNF-alpha, Il-1beta, nitric oxide, and oxygen radicals in THP-1 cells, and induces cell apoptosis (Figure 1). Additionally, it causes activation of NF-κB, which triggers a cascade of reactions leading to the secretion of pro-inflammatory cytokines and cell apoptosis [32]. The VacA cytotoxin also depolarizes the cell membrane, alters mitochondrial membrane permeability, disrupts endosome and lysosome function, activates tumor process kinases, and inhibits antigen presentation and T cell activity [33,34].

### 2.2. H. pylori and Host Factors in Gastric Lymphogenesis

Research conducted in recent years has demonstrated that *H. pylori* colonization is crucial in the early phase of lymphogenesis (Figure 2) [35,36,37]. Chronic *H. pylori* infection induces the secretion of pro-inflammatory cytokines by macrophages and dendritic cells, such as interleukin-1 beta (IL-1β), TNF-α, IL-8 and IL-6, and allows differentiation of Th1 and Th17 cells, which are involved in establishing persistent inflammation [38]. In gastric MALT-lymphoma, *H. pylori*-induced antigenic stimulation results in the formation of T-cell infiltrates that invade and destroy gastric glands [39]. These reactive Th1 CD4^+^ lymphocytes produce high levels of IFN-γ, and facilitate the proliferation of neoplastic B cells through CD40L-CD40 co-stimulation, and the secretion of Th2 cytokines including IL-4 [40]. The majority of CD4^+^ T-cells are suppressive CD25^+^ forkhead box P3 (FOXP3)^+^ regulatory T-cells (T_regs_), which are themselves recruited by tumor B-cells; it has been suggested that higher numbers of tumor-infiltrating FOXP3^+^ cells are associated with better response to *H. pylori* eradiation therapy [41]. Recent findings indicated that a proliferation-inducing ligand (APRIL) expressed by neutrophils, eosinophils and tumor-infiltrating macrophages seems to be important for gastric lymphogenesis induced by *H. pylori* [42]. Recently, Blosse et al. characterized the inflammatory response associated with gastric MALT lymphoma in the stomach of transgenic C57BL/6 mice, and additionally found APRIL-producing eosinophilic polynuclear cells in the lymphoid infiltrates of patients with gastric MALT lymphoma [39]. The authors have demonstrated that the T_reg_-balanced inflammatory environment is an important contributor to gastric lymphogenesis [39]. In recent years, studies have examined the role of the PD-1 pathway in the context of *H. pylori* colonization, and T-cell dysfunction and PD-1 expression have been observed in these patients [43]. High levels of PD-L1 were found in human gastric biopsies taken from patients infected with *H. pylori,* when compared to *H. pylori* negative controls [44]. Shen et al. have demonstrated that the PD-1/PD-L1 checkpoint is involved in intraepithelial neoplasia and early-stage gastric cancer [45]. Furthermore, Holokai et al. hypothesized that *H. pylori*-induced PD-L1 expression within the gastric epithelium is mediated by the Shh signaling pathway during infection. The authors demonstrated that metaplastic cells may survive chronic inflammation by expressing the immunosuppressive ligand PD-L1, which would account for the persistence of the infection and progression to cancer [46]. The role of *H. pylori*-induced PD-L1 expression in lymphogenesis needs to be determined, especially in the context of new therapeutic options such as PD-1 and PD-L1 inhibitors, that act to control the immune checkpoints.

Previous observations on lymphogenesis have shown that host factors may also be important in the development of MALT lymphomas. It has been demonstrated that polymorphisms in TNF-α, GSTT1 and CTLA4 genes are associated with the risk of gastric MALT lymphoma [26]. Moreover, polymorphism of IL-22 was also associated with the susceptibility to gastric MALT lymphoma (Figure 2) [47]. Liao demonstrated that the C allele at rs1179246, C allele at rs2227485, A allele at rs4913428, A allele at rs1026788 and T allele at rs7314777 were significantly associated with increased risks of the disease, and the pattern of IL-22 expression in gastric mucosa predicted treatment responses to *H. pylori* eradication in patients with *H. pylori*-induced gastric MALT lymphoma [47]. These findings suggest that host factors and their alterations may be crucial for the pathogenesis of MALT lymphomas.

### 2.3. The Role of Epigenetic Factors and Molecular Factors in Gastric Lymphogenesis

The mechanisms underlying the progression of gastric MALT lymphomas are less clear, and are currently the subject of intensive research. It has been suggested that in later phases of the disease process there are some other factors, such as host, genetic and molecular factors, as well as changes in the tumor microenvironment, making the role of *H. pylori* is less relevant than during the initial stages of the disease [26,48]. Tumor progression is now known to be driven by an interaction between B-cell receptor (BCR)-derived signals and T-helper (T_h_) cell signals [42]. Moreover, it has been shown that chemokine receptors play a crucial role in malignant B-cell migration and transformation. It has been has found that the expression of the chemokine receptor, CXCR3 and its ligand Mig, which is expressed on activated T-cells and malignant B-cells, may be correlated with the metastatic migration of neoplastic B-cells into other organs. It has been suggested that this migration may indicate a loss of dependence on *H. pylori* and progression to advanced-stage MALT lymphoma [26]. Deutsch et al. further suggested that the development of gastric MALT lymphoma is associated with increased CCR7, CXCR3 and CXCR7 expression and a loss of CXCR4. Transformation of gastric MALT lymphoma to extranodal DLBCL was accompanied by significant upregulation of chemokine receptors CCR1, CCR5, CCR8, CCR9, CXCR6, CXCR7 and XCR1 [49].

Epigenetic and genetic changes are also involved in lymphogenesis. Recent reports in murine models have revealed that micro RNA (miR), small non-coding RNA molecules play important roles in the regulation of cellular proliferation and apoptosis. Blosse et al. have demonstrated that miR-155, miR-150, miR-196a and miR-138 are upregulated, whereas miR-7 and miR-153 are downregulated in gastric lymphomagenesis in human samples compared to gastritis control samples [50]. In addition, it was found that the expression of miR-203 and its target ABL1 was dysregulated in MALT lymphoma biopsy samples [51]. miR-155 deserves special attention, as it plays a crucial role in the regulation of inflammation and immune responses. In vivo animal experiments have shown that miR-155 is necessary to control *H. pylori* infection through Th1 and Th17 responses contributing to bacterial persistence [50]. Furthermore, it has been demonstrated to be expressed at much higher levels in *H. pylori*-independent tumors than in *H. pylori*-dependent tumors [26,52]. These findings indicate that miRs are important molecules in the tumor environment, however further studies are needed in order to clarify the exact mechanism of miR dysregulation in lymphogenesis. Other studies highlight that epigenetic modifications through chromatin and DNA methylation are key events in the progression of the early stages of MALT lymphoma [26]. It was demonstrated that *H. pylori* causes DNA methylation and hypermethylation of the CpG islands, which result in the deletion and loss of expression of tumor suppressor genes [53]. Park et al. have demonstrated that the methylation of the cyclin-dependent kinase inhibitor p16^INK4A^ occurred in 75% of patients with gastric MALT lymphoma and it was more frequently in t (11:18) (q21;q21)-negative gastric MALT lymphomas [54]. Aberrant CpG methylation within certain genes, such as p16, MGMT and MINT31, was associated with *H. pylori* infection [26].

In light of the above, it cannot be excluded that other species of the *Helicobacter* genus, such as *Helicobacter heilmannii*, and gut microbiota interactions with *H. pylori* do not play important roles in lymphogenesis [55,56]. Recent studies have focused on intestinal microbes that may have impacts on local and distant tumor formation through disturbances in the ratio of its components, infection, microbial products or by modulating tumor immunosurveillance, however, the exact role of microbiota is not entirely clear [57,58].

### 2.4. H. pylori Eradication in B-Cell Lymphomas

Based on ESMO guidelines and Maastricht consensus recommendations, eradication of *H. pylori* infection using a combination of antibiotics and proton-pump inhibitors (PPI) for 7 days is the first-line treatment of gastric MALT lymphoma [59,60]. Epidemiological and clinical studies conducted over a number of years demonstrated regression of infiltrative lesions following *H. pylori* eradication in 60–90% of patients with low-grade gastric MALT lymphomas or early-stage DLBCL [20,61,62]. Based on clinical observations, eradication therapy is usually recommended for *H. pylori*-positive lymphomas but is also indicated in *H. pylori*-negative lymphomas, showing response rates of 83% [19,60,62]. Regression of submucosa localized lymphoma infiltrates has been reported in more than 80% of cases, whereas regression was present in around 50% of cases with deeper invasion [63]. In a study conducted by Moleiro et al., it was demonstrated that relapse occurred in 14% of the patients after a mean period of 21 months [64]. The authors indicated that relapse rates were higher in patients with *H. pylori* re-infection, in cases where more than one eradication regimen was used, and in cases with lymphomas localized in the corpus [64]. Based on the latest Maastricht consensus as the first line of empirical therapy, bismuth-based quadruple therapy or concomitant quadruple therapy can be used [60]. Due to insufficient numbers of prospective clinical studies, the optimal therapy regimens for gastric lymphomas with or without *H. pylori* infection have not been determined [19,65]. Other treatment options, including radiotherapy, chemotherapy and immunotherapy should be considered in all patients who do not respond to eradication therapy, or in patients with *H. pylori*-negative lymphomas. Moreover, similar recommendations should be made for patients with the presence of tumors housing the t (11;18) translocation that are less susceptible to eradication treatment. An emerging problem in all regions of the world is the increasing *H. pylori* antibiotic resistance, particularly to clarithromycin. This means that clarithromycin-based treatment cannot be considered without a further testing regime to see whether the organisms are sensitive [66,67]. Resistance of *H. pylori* strains to various antibiotics, mainly clarithromycin and levofloxacin, may be the one of the reasons for treatment failure.

There are some findings demonstrating that the lack of tumor regression after eradication therapy may be related to a complex process of multiple changes in the tumor immunological environment, such as lack of macrophage activity, upregulated expression of p-SHP, p-ERK, and Bcl-XL, nuclear translocation of NFATc1, and CD56+ NK cell activity. The member of the nuclear factor of the activated T cell family, NFATc1 has been detected in B-cell lymphomas, such as Burkitt lymphoma, Hodgkin lymphoma, and MALT lymphoma and it is supposed to be involved in lymphogenesis [26].

### 2.5. H. pylori and Gastric DLBCL

The association between *H. pylori* infection and DLBCL localized in the gastrointestinal tract has been theorized for a long time. DLBCL is the most common subtype of NHL and comprises a heterogeneous group of tumors. Among extra-nodal manifestations, primary gastrointestinal lymphomas are the most common presentation [63,68]. In some cases, MALT lymphoma may gradually progress to an aggressive DLBCL. For many years it was believed that MALT lymphoma is *H. pylori*-dependent and that this is lost with a high-grade transformation, but many reports published so far have demonstrated *H. pylori* dependence in both low-grade and high-grade gastric MALT lymphomas [28,29,69]. According to the WHO classification from 2016, it is recommended that when gastric MALT lymphoma patients demonstrate transformation into large-cell lymphoma, they should be re-classified as DLBCL; a distinct classification from de novo DLBCL, without histological infiltration of centrocyte-like cells in the lamina propria, and typical lympho-epithelial lesions [3,63,68]. *H. pylori* eradication is not a standard treatment for primary gastric DLBCL, however, some prospective studies have shown the regression of *H. pylori*-positive early stage (Lugano stage IE or II1) gastric DLBCL lymphoma after *H. pylori* eradication and complete response in two-thirds of patients [70]. Moreover, it was demonstrated by Kuo et al. that in all patients with *H. pylori*-positive DLBCL successfully treated with first-line eradication treatment, the overall survival was significantly higher than in *H. pylori*-negative DLBCL cases (76.1% vs. 39.8%, *p* < 0.001) [71]. Interestingly, it was suggested that *H. pylori*-dependent gastric DLBCL possessed better overall survival than *H. pylori*-independent cases [71]. In the study published recently, Ben Younes et al. highlighted the role of the AKT signaling pathway in *H. pylori*-induced tumorigenesis and progression of MALT lymphoma into DLBCL [68]. The authors indicated that CagA localization in B-cells may predispose them to accumulate multiple genetic and epigenetic changes leading to loss of PTEN and/or cyclin A2 overexpression which were significantly associated with consecutive AKT activation [68]. Moreover, recent findings reported by Tsai et al. further demonstrate that most *H. pylori*-dependent tumors express CagA and BCL10, and CagA expression in post-*H. pylori* eradication biopsies is downregulated [70]. The authors pointed out that immunohistochemical assays for nuclear BCL-10 or NK-κB (p65) and CagA expression can help to predict *H. pylori* dependence in patients with early-stage gastric MALT lymphoma and gastric DLBCL (MALT) who receive first-line eradication treatment [70]. Researchers have also shown that some patients diagnosed with de novo DLBCL achieved complete remission after *H. pylori* eradication [69]. Moreover, Cheng et al. have demonstrated that positive *H. pylori* status was associated with better prognosis in patients with gastric de novo DLBCL [72]. Nevertheless, DLBCL is an aggressive tumor that may progress rapidly, and in some cases, relapses or progression after eradication treatment have been reported. This highlights that further studies are needed to help to stratify the group of patients with the risk of progression, to establish when *H. pylori* eradication therapy can be safely recommended and when and in which group of patients other treatment strategies should be implemented.

### 2.6. H. pylori and Burkitt Lymphoma

There are few cases in the literature, presented mainly as case reports, which demonstrate patients with gastric Burkitt lymphoma associated with *H. pylori* infection [73,74,75,76,77]. Burkitt lymphoma is a very aggressive type of NHL that is commonly localized in the gastrointestinal tract, although rarely in the stomach. While gastric Burkitt lymphoma in adults has a very low incidence, in children, its occurrence is even rarer. Most reports on the relationship between *H. pylori* and gastric Burkitt lymphoma presented cases of patients with a tumor mass primarily restricted to the stomach [74,75,76]. The possible role of *H. pylori* infection in the pathogenesis of Burkitt lymphoma pathogenesis needs to be evaluated, and some authors have speculated of a possible link related to the host immune response to CagA or other *H. pylori* virulence factors [76]. Another hypothesis is that gastric Burkitt lymphoma shares common developmental pathways with MALT lymphoma of the stomach [77]. The relationship between plasmocytomas and *H. pylori*, and more specifically the disappearance of these tumors after eradication treatment has also been reported [78].

### 2.7. Campylobacter jejuni and IPSID

*Campylobacter jejuni* (*C. jejuni*) is a microaerophilic Gram-negative bacillus that is responsible for asymptomatic carrier and mild, self-limiting gastroenteritis. However, more severe infections can lead to sepsis, and it is a known initiating agent of autoimmune diseases such as Guillain–Barre syndrome and reactive arthritis. Infection is quite common in humans worldwide and is associated with the consumption of contaminated food, mainly poultry and dairy products. Chronic antigenic stimulation in the course of campylobacteriosis has been linked to immunoproliferative small intestine lymphoma (IPSID), also known as alpha chain disease (ACD) [19,53]. This disease is classified as a variant of MALT which is localized primarily in the small intestine, but can be also detected in the stomach, colon, rectum, mesenteric lymph nodes, as well as other organs [79]. The disease has been described in young adults from the developing world, including India, the Mediterranean Basin, the Middle East, Africa, and South America, where low socioeconomic status, poor sanitation, malnutrition, and frequent enteric infections are common [80]. The disease is characterized by lymphoplasmocytic intestinal infiltrates with monotypic α-heavy chain expression. The pathogenesis is poorly understood and the association of *C. jejuni* infection with the development of infiltrative lesions is based on the detection of the genetic material of these bacteria in tissue from IPSID patients [81,82]. Moreover, clinical observations have shown the regression of infiltrative lesions in the course of IPSID, mainly from the macrolide or tetracycline group combined with antiparasitic drugs [81]. Nevertheless, the disease may progress from early plasmocytic lesions of low malignancy to a high-stage immunoblastic disease [19,80]. Cases of plasmoblastic lymphoma and DLBCL as progressions of IPSID have also been described [80,83]. The exact mechanism by which *C. jejuni* may contribute to the development of IPSID lesions is not clear. Some authors have hypothesized a role for *C. jejuni* CDT cytotoxin. It can be speculated that, as in the case of *H. pylori*, CDT-vacuolating cytotoxin enables the destruction of intestinal villi and invasion of the intestinal mucosa. Furthermore, CDT toxin leads to DNA damage as demonstrated in mouse models [84]. It has been demonstrated that cells exposed to CDT-DNA damaging agents suffer extensive genetic modifications that could cause apoptosis; hence, researchers speculate that the human clinical isolate *C. jejuni* 81–176 is able to promote colorectal cancer [85]. Adhesion of bacteria to endothelial cells in most cases leads to a strong immune response. However, it seems that, similar to *H. pylori*, *Campylobacter* infection can lead to asymptomatic fecal carriage in immunocompromized hosts [82]. The role of *C. jejuni* in intestinal lymphogenesis requires further investigation, along with a host of other pathogenic bacteria associated with the development of IPSID, such as *Campylobacter coli*, *H. pylori*, *Vibrio fluvialis*, *Escherichia coli* [80].

## 3. Role of Bacteria in Skin and Ocular NHL Lymphomas

### 3.1. Borrelia burgdorferi and Cutaneous NHL

Primary cutaneous NHL represent the second most common location of NHL after lymphomas in the gastrointestinal tract, with B-cell lymphomas accounting for approximately 30% of all cutaneous lymphomas. The association of *Borrelia burgdorferi* infection with the development of indolent lymphomas such as cutaneous MZL and follicular lymphoma (FL), but also aggressive lymphomas such as DLBCL with cutaneous localization and Mantle cell lymphoma has been described [86]. *Borrelia burgodorferi* is a spiral bacterium responsible for Lyme disease and the spirochetes are transmitted by *Ixodes damini* ticks. Chronic infection in the course of Lyme borreliosis such as acrodermatitis chronica atrophica, and polyradiculoneuritis may lead to the development of B-cell lymphomas. The DNA of *Borrelia burgdorferi* has been detected in biopsy samples of patients from Europe and Australia diagnosed with cutaneous MALT lymphomas and also in tissue samples from patients diagnosed with FL and DLBCL [14,53]. Furthermore, clinical observations from case reports of patients with low-stage cutaneous MALT lymphoma treated with antibiotics alone, mainly from the cephalosporin and tetracycline groups, led to regression of infiltrative lesions [5,86]. Based on these reports, therapy with oral antibiotics is currently acceptable as first-line treatment [87]. Travaglino et al., in a meta-analysis, showed that *Borrelia burgdorferi* was significantly associated with primary cutaneous lymphoma in endemic areas such as North American and Europe (from southern Scandinavia into the northern Mediterranean countries of Italy, Spain, and Greece, east from the British Isles into central Russia and the northeastern and north-central United States) [86,88]. It seems that in the course of local *Borrelia* infection, atypical lymphoid follicles are formed in the skin, and lymphocytes may further infiltrate the dermis and produce borrelial “lymphocytoma” which can be difficult to distinguish histologically from MZL. There are data demonstrating that persistent inflammation in the course of *Borrelia* infection may lead to monoclonal B-cell proliferation and BCL-2 protein expression [89,90]. Cutenous MALT may be associated with *Borrelia afzelii*, following the presentation of a case report [91]. However, the role of other *Borrelia* species in the development of B-cell lymphoma is still controversial.

### 3.2. Chlamydophila psittaci (Ch. psittaci) and Ocular MALT

Another example of the relationship between the inflammatory process induced by intracellular bacteria and lymphogenesis is the association of *Chlamydophila* species with ocular adnexal lymphoma. Ocular adnexal MALT lymphomas account for approximately 5–15% of all MALT lymphomas and their incidence is increasing in recent years [92]. The prevalence is highest in elderly patients older than 65 years of age, living in rural areas with a history of chronic conjunctivitis [53]. The lymphoma lesions occur principally in the conjunctiva, orbital soft tissue, and lachrymal apparatus, with bilateral involvement in 10–15% of cases without ocular infiltration [53,93]. The etiology of ocular adnexal MALT lymphoma is currently unknown, and observations have yielded conflicting results. Some reports demonstrated the presence of *Ch. psittaci* DNA in 11–87% of biopsy samples from patients in different geographical regions [92,93,94]. However, and conflicting with the previous research, Zang et al. did not confirm the association between *Ch. psittaci* and ocular lymphomas [95]. In a recent meta-analysis, Travaglino et al. showed that the prevalence of *Chlamydia* in patients with lymphoma varies widely, being most common in Korea and Italy [92]. Furthermore, these authors demonstrated that not only was *Ch. psittaci* detected in tissues in the course of ocular adnexal lymphoma but also *Ch. pneumoniae* in patients in China and *Ch. trachomatis* in patients in Great Britain [92]. It has been speculated that *Chlamydia* may play a role in the development of MALT lymphoma of other sites, such as the lung, skin, uterus, bowel, and stomach [92,93,96]. *Chlamydia* was isolated from conjunctival swabs and from blood samples taken from patients with lymphoma [14,97]. *Chlamydia* is intracellular bacteria that infect humans through contact with infected birds, most commonly leading to asymptomatic infections, but can also cause chronic conjunctivitis, pneumonia, hepatitis, and pericarditis [14,94]. The properties of *Chlamydia* that allow it to establish persistent infections are related to their complex developmental cycle and their occurrence in three forms: elementary body (EB), which is its metabolically inactive infectious form, metabolically active intracellular growth stage form called the reticulate body (RB) and intermediate body (IB). The ability of the bacteria to modify their life cycle in response to a changing environment leads to resistance of the infected cell to apoptosis [97,98]. However, a link between chlamydial life cycle and lymphogenesis needs to be established. Persistent infections that induced polyclonal B-cell expansion and proliferation were evaluated by detection of somatically hypermutated immunoglobulin genes with an ongoing mutations pattern. Moreover, chronic antigenic stimulation may lead to chromosomal abnormalities with genetic and epigenetic alterations resulting in activation of the NF-κB pathway [98,99]. Persistent infection leads to the formation of cells that gradually become independent of their involvement in the microenvironment. Chronic stimulation due to *Ch. psittaci* infection may be favored by molecular mimicry, as *Ch. psittaci* is able to induce immune reactions that cross-react with the host self-antigens, leading to a failure to eliminate the pathogen and induce lymphogenesis [26,98]. Furthermore, the progression of OAMZL to the more aggressive DLBCL may be independent of chronic antigenic stimulation provided by the microorganism and instead be induced by mutations of tumor suppressor genes such as p53 and p16 [26,98,99].

Considering the fact that both the type of lymphoma and *Ch. psittaci* infection are rare diseases, there is currently no universal recommendation regarding the therapeutic approach. Often the lesions are located superficially, indolent and rarely lead to progression to more malignant types of lymphoma. Regression of lesions after antibiotic therapy with doxycycline was observed in 65% of patients in randomized control trials phase II [100]. Other treatment options include surgery and observation, radiotherapy, immunotherapy, radioimmunotherapy and immunomodulating agents in relapsed cases [97,98].

## 4. Role of Bacteria in Pulmonary Lymphogenesis

Pulmonary MALT-tissue lymphoma also known as bronchial-associated lymphoid tissue (BALT) lymphoma, is the most common B-cell lymphoma in lungs, and the prevalence of this type of disease accounts for 7–8% of all MALT types [101]. Considering that chronic inflammatory and autoimmune disorders play an important role in the pathogenesis of these lymphomas, a link between various micro-organisms and pulmonary MALT lymphoma has been sought, but so far the clear link with a particular infectious agent has not been identified. This disease is characterized by a slow-growing tumor, which infiltrates epithelial tissue and forms lympho-epithelial lesions. The disease, in most cases and as in other localizations of extranodal MALT lymphomas, is limited (IE or IIE) and a specific but relatively rare cytogenetic aberration for MALT lymphomas is t (11;18) (q21;q21), which is found in approximately 30–50% of MALT lymphomas with gastrointestinal and pulmonary localizations [101].

### Achromobacter xylosoxidans and BALT

*Achromobacter xylosoxidans* is one candidate pathogen associated with pulmonary MALT, however recent work on the association of this bacterium with lymphogenesis brings conflicting results. Adam et al. examined lung tissue from 124 European patients with pulmonary MALT and genetic material of a Gram-negative rod from the *Alcaligenes* family was detected in 46% of patients vs. 18% in controls, with a significant difference in the prevalence rate for the different geographic regions, ranging from 33% to 67% [102]. In another study from a Japanese cohort analyzing tissue samples from 52 patients with pulmonary MALT and 18 patients of pulmonary DLBCL, the presence of *Achromobacter xylosoxidans* DNA was found only in 11% of DLBCL cases and 2% of BALT cases. The authors of the second paper speculated that differences in the prevalence of this bacterium between Europe and Asia may significantly affect the results of the analysis. Moreover, the diagnosis of pulmonary MALT itself is an important issue and may be difficult as it requires histopathological confirmation of clonal proliferation in all BALT cases to differentiate it from reactive lymphoid hyperplasia [103]. *Achromobacter xylosoxidans* is an opportunistic bacterium, with a low virulent potential although is very frequently resistant to antibiotics. The pathogen is usually isolated from patients with cystic fibrosis [104]. Clinical manifestations of infection are seen mainly in immunocompromised patients and include pneumonia, urinary tract infections, meningitis, and sepsis [104]. The question of whether this bacterium will be like *H. pylori* in gastric MALT is still open. In contrast, a recent metagenomic study by Borie et al. found no evidence that any bacterial, fungal, viral or parasitic pathogen is associated with pulmonary MALT [105].

The association between BALT and other bacterial pathogens such as Mycobacterium tuberculosis, Mycobacterium avium, Chlamydophila pneumoniae, Chlamydia trachomatis, Chlamydophila psittaci and Mycoplasma pneumoniae has been suggested [93,106,107]. Two cases of successful antibiotic therapy with clarithromycin for the treatment of BALT are described [108]. The role of infections in the development of BALT remains unclear. Some authors suggest the role of some pre-existing autoimmune disorders such as Sjögren’s syndrome, reactive arthritis or Hashimoto’s thryroiditis. Because of the rarity of these types of lymphoma and the indolent clinical course, the optimal treatment strategies are not well defined and include surgery, radiation, chemotherapy and immunotherapy [109,110].

## 5. *Escherichia coli* and Primary Bladder MALT

Another pathogen that may also be associated with MALT lymphomas is *Escherichia coli* (*E. coli*). Recurrent infections with these bacteria have been reported in patients with primary bladder MALT lymphoma, an exceedingly rare form of lymphoma [111]. Based on the literature, 58 patients with this type of lymphoma were found, mainly from Asia and the UK with female predominance. A case of MALT lymphoma in the stomach and bladder was also described [111,112]. Chronic, recurrent cystitis of *E. coli* etiology has been observed in approximately 30% of patients, and it can be speculated that persistent cystitis is a necessary precursor of lymphoma. There are no treatment guidelines, with surgical excision, chemotherapy, radiation, or combined modality being utilized [111,113]. Furthermore, some patients were successfully treated with antibiotics [111,114,115]. *E. coli* is a Gram-negative rod-shaped bacterium. *E.coli* harmless strains are part of the natural bacterial microflora of the human body and they colonize the human digestive tract but there are also pathogenic *E. coli* strains responsible for many infections, including those of the urinary tract infections, wounds, pneumonia, diarrhea, meningitis and sepsis. Uropathogenic *E. coli* (UPEC) strains can permanently colonize the urinary tract, leading to an acute inflammatory process in about 70% of cases. More importantly, these strains can also lead to recurrent and persistent infections [116]. It is not entirely clear why some strains can invade the bladder mucosa and others cannot, and further research is needed on the exact mechanism of complex interactions of *E. coli* with host cells. In vitro studies have shown that UPEC *E. coli* strains through virulence factors, mainly adhesins, toxins, complex iron-uptake systems, and immune evasion strategies, have the ability to persist intracellularly in bone marrow-derived macrophages and uroepithelial cells and establish long-term infection [116,117]. UPEC are able to exist in intracellular bacteria communities, and inhibit the immune response by actively blocking TLR-4 signaling, NF-κB activity and pro-inflammatory cytokine production in urothelial cells [117]. However, the exact mechanisms of *E. coli’s* role in lymphogenesis require further study. Given that *E. coli* infection is common worldwide, as is *H. pylori*, whereas the incidence of primary bladder MALT lymphoma is very rare, the role of this bacterium in the formation of bladder MALT lymphoma requires further investigation.

## 6. *Coxiella burnetii* (*C. burnetii*) and Various NHL

The list of bacterial pathogens involved in lymphogenesis has also recently included *C. burnetii*, the Gram-negative intracellular pathogen responsible for Q fever [118]. The role of this bacterium in the development of DLBCL and other NHL lymphomas was first reported in 2016, when *C. burnetii* lymphadenitis and hemophogocytic syndrome was linked to NHL lymphomas, such as DLBCL, MALT-lymphoma, FL, Mantle cell lymphoma and chronic lymphocytic leukemia [14]. *C. burnetii* infection is primarily zoonotic, acquired by respiratory droplets and is mainly asymptomatic in humans, however acute infection may manifest as flu-like illness, pneumonia, hepatitis, and a small percentage of patients may develop persistent infection which manifests as endocarditis, vasculitis, and lymphadenitis. *C. burnetii* was detected in monocytes, macrophages and dendritic cells in both chronic lesions such as granulomas and in NHL lesions. The role of *C. burnetii* in the pathogenesis of NHL is not fully understood. *C. burnetii* DNA was present in about 36% of cases with NHL vs. 8% of controls but this difference was not statistically significant [118]. It has been speculated that the risk of NHL could be increased after exposure to *C. burnetii*. Research into the relationship between this infection and NHL has been conducted in the Netherlands, where an outbreak of Q fever was reported between 2007 and 2010. The median time between primary *C. burnetii* infection and NHL diagnosis was 8 months, and the most common forms of lymphoma were CLL. Moreover, Melanotte et al. reported that Q fever most commonly preceded DLBCL and mantle cell lymphoma [118,119]. According to the latest data, 45 cases of NHL associated with *C. burnetii* persistent infection have been described so far [14]. It has been reported that persistent Q fever is associated with an altered Th1 response with defective production of IFN-γ and overproduction of proinflammatory cytokines IL-10 and IL-6 [120]. The authors demonstrated in murine models that, in the absence of T-bet, which is a transcription factor known to initiate and coordinate the gene expression program during Th1 differentiation and is crucial for clearance of intracellular pathogens, leads to defective bacterial control, persistent infection, and organ injury manifesting as an increased number of granulomas [120]. The same authors in another study showed that during the course of Q fever soluble E-cadherin (sE-cad) can be detected in the sera of patients, indicating that sE-cad can be considered as a marker of a metabolic disorder and/or bacterial invasion in the course of Q fever. The role of sE-cad as a tumorigenic co-factor was highlighted in the infections caused by *H. pylori*, that trigger gastric adenocarcinoma. The association between sE-cad release and the induction of NHL is unknown so far [120]. In another study conducted by Melanotte et al., it was demonstrated that patients with *C*. *burnetii* lymphadenitis presented significantly elevated levels of *BCL2* and *ETS1* mRNAs, which may indicate the upregulation of anti-apoptotic processes and the fact that lymphadenitis might constitute a critical step towards lymphomagenesis [121]. The association of Q fever with NHL requires further study.

## 7. Conclusions

The involvement of various pathogens in the development of NHL remains an open topic. We cannot forget the proven role of other pathogens such as HIV, HCV, HTLV-1, and EBV which have a known oncogenic potential in the development of NHL (primary splenic MALT lymphoma, Burkitt lymphoma, DLBCL and others). Moreover, a parasite such as *Plasmodium falciparum* is also considered a co-factor with EBV in the development of Burkitt lymphoma. Studies on the role of infectious agents in the lymphogenesis of certain B-cell lymphomas show that diagnosis of these infections, particularly caused by *H. pylori*, *C. jejuni*, *Borrelia burgdorferi* and *Chlamydophila psittaci* among others, antibiotic therapy should be included in the diagnostic and treatment process (Table 1).

On the other hand, it is also necessary to consider the role of other infectious agents, especially viruses in lymphogenesis. In addition, host and tumor genomics and in vitro studies will be important to identify other factors regarding the pathogenesis of lymphoma development and progression, such as novel prognostic markers, microbiota–host relationships and genetic factors affecting the tumor microenvironment (Figure 3).

These important research challenges will be key to identifying new strategies for the treatment and prevention of B-cell lymphomas in the future.

## Figures and Tables

**Figure 1 ijms-22-07372-f001:**
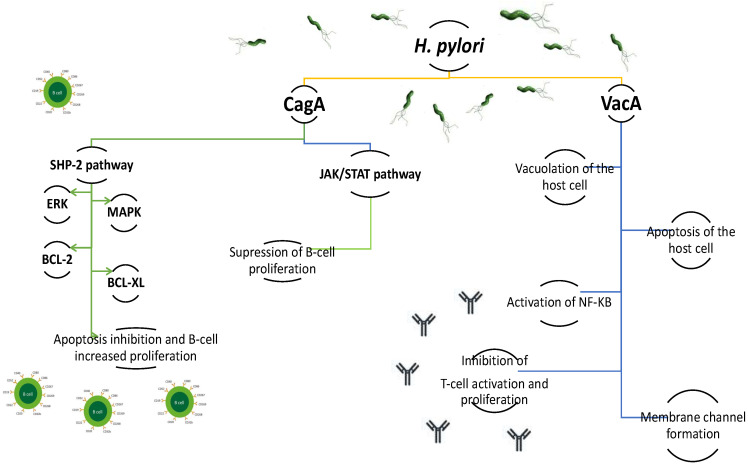
The effect of *Helicobacter pylori* CagA protein on different signaling pathways in the host cells and pathological changes caused by VacA toxin in the host cells. HP-2: Src homology-2 domain phosphatase; ERK: endoplasmic reticulum kinase; MAPK: p38 mitogen-activated protein kinase; BCL-2: B cell lymphoma-2 protein; BCL-XL: B cell lymphoma- XL protein; JAK/STAT pathway: Janus Kinase/Signal Transducer and Activator of Transcription pathway; NFKB: nuclear factor kappa B.

**Figure 2 ijms-22-07372-f002:**
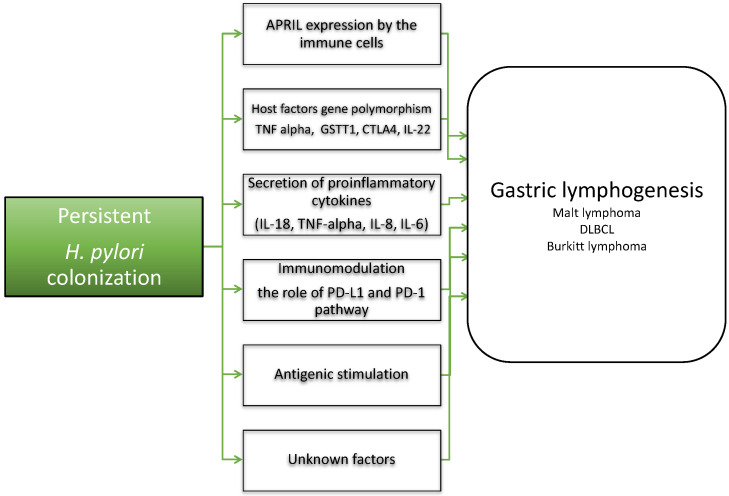
Involvement of *Helicobacter pylori* in gastric lymphogenesis. The complex relationship between pathogen-related events, host factors and immune mechanisms plays a key role in the development of gastric lymphoma. APRIL: a proliferation-inducing ligand; TNF alpha: Tumor Necrosis Factor α; GSTT1: glutathione s-transferases theta 1, CTLA4: Cytotoxic T-lymphocyte-associated protein 4; IL-22: interleukin-22; IL-18: interleukin-18; IL-8: interleukin-8; IL-6: interleukin-6; PD-L1: programmed death-ligand 1; PD-1: Programmed cell death protein 1; MALT: mucosa-associated lymphoid tissue lymphoma; DLBCL: diffuse large B-cell lymphoma.

**Figure 3 ijms-22-07372-f003:**
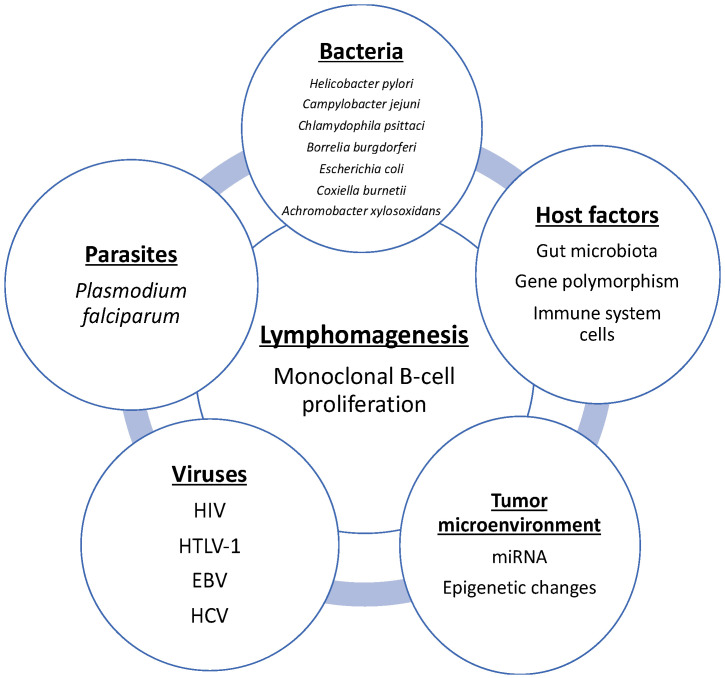
Known factors that play an important role in lymphomagenesis.

**Table 1 ijms-22-07372-t001:** The association of certain bacterial pathogens with B-cell non-Hodgkin Lymphoma.

Pathogen	Type of Lymphoma	Localization
*Helicobacter pylori*	Gastric MALT lymphoma, EMZL, non-gastric MALT DLBCL lymphoma Burkitt lymphoma IPSID	Stomach, intestine
*Campylobacter jejuni*	IPSID	Intestine
*Borrelia burgdorferi,* *Borrelia afzelii*	MALT lymphoma	Skin
*Chlamydophila psittaci*	Ocular adnexal MALT lymphoma	Eye
*Achromobacter xylosoxidans*	BALT lymphoma	Lungs
*Chlamydophila pneumoniae, Chlamydia trachomatis, Mycoplasma pneumoniae,*	BALT lymphoma	Lungs
*Mycobacterium tuberculosis, Mycobacterium avium*	MALT lymphoma	Lungs
*Eschericha coli*	MALT lymphoma	Bladder
*Coxiella burnetii*	MALT lymphoma, DLBCL, FL, B-CLL	Various localizations

MALT: mucosa-associated lymphoma tissue; EMZL: extranodal marginal-zone lymphoma, DLBCL: diffuse large B-cell lymphoma, IPSID: immunoproliferative small intestine lymphoma, BALT: bronchial-associated lymphoid tissue lymphoma, FL-follicular lymphoma, B-CLL: B-cell chronic lymphocytic leukemia.

## Data Availability

Not applicable.

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
