# Peer review of "Bacterial Infection and Non-Hodgkin B-Cell Lymphoma: Interactions between Pathogen, Host and the Tumor Environment"

_ijms, 2021, doi:10.3390/ijms22147372_

Round 1

Reviewer 1 Report

Article is very interesting and important. Role of infections in cancer development is very significant.

1. Authors can in Introduction describe, that bacteria have impact also on other cancer types https://www.sciencedirect.com/science/article/abs/pii/S0024320520301454 https://www.mdpi.com/2076-2607/7/1/20 https://journals.viamedica.pl/folia_histochemica_cytobiologica/article/view/FHC.2011.0036/3377. 

2. Authors presented mechanisms of action, but there are not drawings, which would facilitate a look at the mechanisms of action and the proteins and pathways involved. Authors must add some figures, that will make the article excellent.

Author Response

We are sincerely grateful to the reviewers for their effort and time invested in evaluating our manuscript. Their insightful comments and suggestions helped us greatly improve our manuscript. All major changes in the revised manuscript are marked in red.

Response to Reviewer 1

Comment 1. Authors can in Introduction describe, that bacteria have impact also on other cancer types https://www.sciencedirect.com/science/article/abs/pii/S0024320520301454 https://www.mdpi.com/2076-2607/7/1/20 https://journals.viamedica.pl/folia_histochemica_cytobiologica/article/view/FHC.2011.0036/3377. 

Thank you for this suggestion. The following statements were added in the Introduction Section:

Some specific bacterial species have been identified that correlate strongly with cancers. The microorganisms include Salmonella typhimurium which associates with hepatobiliary carcinoma, Citrobacter rodentium that can be a leading cause of human colorectal cancer, Mycobacterium tuberculosis which associates with lung cancer [ Laliani Life Sciences 2020, Song et al., Infectious Agents and Cancer BioMed Central]] and two periopathogenic species Fusobacterium nucleatum and Porphyromonas gingivalis that play an important role in the development of colorectal and pancreatic cancer.[ Karpiński TM Microorganisms Microorganisms 2019, 7, 20; doi:10.3390/microorganisms7010020] Some research also demonstrated the role of infections caused by bacteria, such as Streptococcus sp., Peptostreptococcus sp., Prevotella sp., Fusobacterium sp., Porphyromonas gingivalis, and Capnocytophaga gingivalis in pathogenesis of the oral cancer. [Karpiński].

In the manuscript the most recent publications suggested by the reviewer were cited [6, 7, 8] with the exception of Rybojad et al (https://journals.viamedica.pl/folia_histochemica_cytobiologica/article/view/FHC.2011.0036/3377), which is more concerned with the etiology of pulmonary infections in patients with lung cancer.

  1. Authors presented mechanisms of action, but there are not drawings, which would facilitate a look at the mechanisms of action and the proteins and pathways involved. Authors must add some figures, that will make the article excellent.

Thank you very much for your remarks. The following drawings were added in the manuscript.

Figure 1. The effect of Helicobacter pylori CagA protein on different signaling pathways in the host cells and pathological changes caused by VacA toxin in the host cells. Based on Melenotte et al. [14] and Kuo et al. [26]. SHP-2: Src homology-2 domain phosphatase; ERK: endoplasmic reticulum kinase; MAPK: p38 mitogen-activated protein kinase; BCL-2: B cell lymphoma-2 protein; BCL-XL: B cell lymphoma- XL protein; JAK/STAT pathway: Janus Kinase/Signal Transducer and Activator of Transcription pathway; NFKB: nuclear factor kappa B.

Figure 2. Involvement of Helicobacter pylori in gastric lymphogenesis. Complex relationship between pathogen-related events, host factors and immune mechanisms plays key role in the development of gastric lymphoma.  APRIL: a proliferation inducing ligand; TNF alpha: Tumor Necrosis Factor α; GSTT1: glutathione s-transferases theta 1 , CTLA4: Cytotoxic T-lymphocyte-associated protein 4; IL-22: interleukin-22; IL-18: interleukin-18; IL-8: interleukin-8; IL-6: interleukin-6; PD-L1: programmed death-ligand 1; PD-1: Programmed cell death protein 1; MALT: mucosa associated lymphoid tissue lymphoma; DLBCL: diffuse large B-cell lymphoma

Reviewer 2 Report

Review

The authors of the publication raised the important issue of the relationship between bacterial infections and the development of non-Hodgkin B-cell Lymphoma. The examples of infections presented in the paper are more and more often referred to in terms of their participation in the development and progression of certain diseases, especially autoimmune and neoplastic diseases. I consider the selection of these pathogens to be very correct. The work contains an exhaustive analysis of the latest reports and research from various regions of the world, which is equally important in such an analysis.

The part of the publication devoted to the participation of Helicobacter pylori is very well prepared, I think that a great value of the publication would be the preparation of a figure showing the signaling pathway described in the part "H. pylori and MALT, DLBCL, and Burkitt lymphoma ”, which will significantly help in understanding the activity of these bacteria.

In the section "Escherichia coli and primary bladder MALT" it is worth mentioning that E. coli is a natural bacterial microflora and the human body is constantly exposed to them. On the other hand, as emphasized by the authors, pathogenic E. coli strains are an equally important factor.

Minor substantive and editorial comments

In line 454-455 the authors mentioned „Chlamydophila trachomatis” among the factors, the correct name according to the taxonomy is Chlamydia trachomatis, but Chmaydophila pneumoniae and Chmlamydophila psittaci. Similarly in Table 1.

In verse 515 the authors write: "... overproduction of immunoregulatory cytokines IL-10 and IL-6", IL- is probably classified as proinflammatory cytokines, although it may also have regulatory functions.

Editorial notes: no italics in the names of microorganisms in lines: 26, 71, 119, 211, 247, 288, 340, 369, 371, 431, 453, 454, 455, 463.

In line 492 the chapter title moved to the right, line 498 - ending brace, no start brace.

The list of cited literature also lacks italics in Latin names of microorganisms.

Summary

The paper sent for review is very well prepared both in terms of content and editorial, it contains a thorough analysis of almost 120 works from the last few years.

The paper is perfectly suitable for publication, it will be of great substantive value for other researchers.

Author Response

We are sincerely grateful to the reviewers for their effort and time invested in evaluating our manuscript. Their insightful comments and suggestions helped us greatly improve our manuscript. All major changes in the revised manuscript are marked in red.

Point 1: The part of the publication devoted to the participation of Helicobacter pylori is very well prepared, I think that a great value of the publication would be the preparation of a figure showing the signaling pathway described in the part "H. pylori and MALT, DLBCL, and Burkitt lymphoma ”, which will significantly help in understanding the activity of these bacteria.

Thank you for this suggestion. The following drawings were added in the manuscript

Figure 1. The effect of Helicobacter pylori CagA protein on different signaling pathways in the host cells and pathological changes caused by VacA toxin in the host cells. Based on Melenotte et al. [14] and Kuo et al. [26]. SHP-2: Src homology-2 domain phosphatase; ERK: endoplasmic reticulum kinase; MAPK: p38 mitogen-activated protein kinase; BCL-2: B cell lymphoma-2 protein; BCL-XL: B cell lymphoma- XL protein; JAK/STAT pathway: Janus Kinase/Signal Transducer and Activator of Transcription pathway; NFKB: nuclear factor kappa B.

Figure 2. Involvement of Helicobacter pylori in gastric lymphogenesis. Complex relationship between pathogen-related events, host factors and immune mechanisms plays key role in the development of gastric lymphoma.  APRIL: a proliferation inducing ligand; TNF alpha: Tumor Necrosis Factor α; GSTT1: glutathione s-transferases theta 1 , CTLA4: Cytotoxic T-lymphocyte-associated protein 4; IL-22: interleukin-22; IL-18: interleukin-18; IL-8: interleukin-8; IL-6: interleukin-6; PD-L1: programmed death-ligand 1; PD-1: Programmed cell death protein 1; MALT: mucosa associated lymphoid tissue lymphoma; DLBCL: diffuse large B-cell lymphoma

Point 2: In the section "Escherichia coli and primary bladder MALT" it is worth mentioning that E. coli is a natural bacterial microflora and the human body is constantly exposed to them. On the other hand, as emphasized by the authors, pathogenic E. coli strains are an equally important factor.

Thank you for this comment. The following sentences have been corrected according to reviewer’s suggestion.

  1. coli is a gram-negative rod-shaped bacterium. E.coli harmless strains are part of the natural bacterial microflora of the human body and they colonize the human digestive tract but there are also pathogenic E. coli strains responsible for many infections, including those of the urinary tract infections, wounds, pneumonia, diarrhea, meningitis and sepsis.

Minor substantive and editorial comments

Point 3: In line 454-455 the authors mentioned „Chlamydophila trachomatis” among the factors, the correct name according to the taxonomy is Chlamydia trachomatis, but Chmaydophila pneumoniae and Chmlamydophila psittaci. Similarly in Table 1.

The mistakes have been corrected according to your suggestion.

Point 4: In verse 515 the authors write: "... overproduction of immunoregulatory cytokines IL-10 and IL-6", IL- is probably classified as proinflammatory cytokines, although it may also have regulatory functions.

Thank you for your comment, we agree with you, especially as we wanted to point out the importance of pro-inflammatory function.  “It has been reported that persistent Q fever is associated with an altered Th1 response with defective production of IFN-γ and an overproduction of proinflammatory cytokines IL-10 and IL-6.”

Point 5: Editorial notes: no italics in the names of microorganisms in lines: 26, 71, 119, 211, 247, 288, 340, 369, 371, 431, 453, 454, 455, 463.

The mistakes have been corrected.

In line 492 the chapter title moved to the right, line 498 - ending brace, no start brace.

The mistakes have been corrected.

The list of cited literature also lacks italics in Latin names of microorganisms.

The mistakes have been corrected.

Round 2

Reviewer 1 Report

Authors significantly corrected manuscript according to reviewer's suggestions. Recently, I recommend article for publication.